# Economic impacts of a glacial period: a thought experiment to assess the disconnect between econometrics and climate sciences

Marie-Noëlle Woillez[1], Gaël Giraud[1,2,3], and Antoine Godin[1,4]

[1]Agence Française de Développement, 5 rue Roland Barthes, 75012 Paris, France
[2]Centre d'Economie de la Sorbonne, Paris 1 University Panthéon-Sorbonne, 106-112 bd. de l'Hôpital, Paris 75013, France
[3]Chair Energy & Prosperity, Institut Louis bachelier, 28 place de la Bourse, 75002 Paris, France
[4]Centre d'Economie de l'Université de Paris Nord, Université Paris 13 – Campus de Villetaneuse 99, avenue Jean-Baptiste Clément, 93430 Villetaneuse, France

**Correspondence:** Marie-Noëlle Woillez (woillezmn@afd.fr)

**Abstract.** Anthropogenic climate change raises growing concerns about its potential catastrophic impacts on both ecosystems and human societies. Yet, several studies on damages induced on the economy by unmitigated global warming have proposed a much less worrying picture of the future, with only a few points decrease in the world GDP per capita by the end of the century, even for a global warming above $+4\,^\circ$C. We consider here two different empirically estimated functions linking GDP growth or GDP level to temperature at the country level and apply them to a global cooling of $-4\,^\circ$C in 2100, corresponding to a return to glacial conditions. We show that the alleged impact on global average GDP per capita runs from -1.8%, if temperature impacts GDP level, to +36%, if the impacts are rather on GDP growth. These results are then compared to the hypothetical environmental conditions faced by humanity, taking the last glacial maximum as a reference. The modeled impacts on the world's GDP appear strongly underestimated given the magnitude of climate and ecological changes recorded for that period. After discussing the weaknesses of the aggregated statistical approach to estimate economic damages, we conclude that, if these functions cannot reasonably be trusted for such a large cooling, they should also not be considered as providing relevant information on potential damages in the case of a warming of similar magnitude, as projected in the case of unabated greenhouse gas emissions.

## 1 Introduction

Since the first IPCC (1990) report, anthropogenic climate change has been the object of large research efforts. Increased knowledge has raised growing concerns about its potential catastrophic impacts on both ecosystems and human societies if greenhouse gas (GHG) emissions continue unmitigated. In addition to the worsening of mean climate conditions in many places, numerous studies emphasize the risks associated to increased frequency and/or magnitude of extreme events (e.g. droughts, heat waves, storms, floods), rising sea level and glaciers melting (Stocker et al., 2013). These risks have drawn

attention to potential catastrophic consequences for the world's economy (Weitzman, 2012; Dietz and Stern, 2015; Bovari et al., 2018). Yet, several other studies on damages induced on the world economy by unmitigated global warming have proposed a much less worrying picture of the future, with economic damages limited to only a few points of the world's GDP[1] (see Tol (2018) for a review), to the extent that some authors could conclude that *"a century of climate change is likely to be no worse than losing a decade of economic growth"* and hence that *"there are bigger problems facing humankind than climate change"* (Tol, 2018, p. 6). Such results seem surprising when compared to the conclusions of the last IPCC report (Stocker et al., 2013) and to various rather alarming publications since then (Hansen et al., 2016; Mora et al., 2017; Steffen et al., 2018; Nolan et al., 2018). Damage functions[2], either based on enumerative approaches or on econometrics (see section 2), and at the heart of many macroeconomic analyses of climate change impacts, have however been heavily criticized for their lack of empirical or theoretical foundations or for their inadequacy to evaluate the impact of climate change outside the calibration range (Pindyck, 2013; Pottier, 2016; Pindyck, 2017; Pezzey, 2019).

Here, we want to further highlight the disconnect between climate sciences and economic damages projected at the global scale, focusing on econometric approaches. To do so, two different strategies could be considered:

– Carefully list all the climate and environmental changes, as described by climate models for the end of the century (including extreme events, sea level rise, etc.) that are not accounted for by such damage functions and show that the damages would be much larger than the projections obtained with these functions, as done by DeFries et al. (2019) for assessments of climate change damages in general. This option would strongly rely on projections from current Earth system models and there are still many uncertainties, especially on potential tipping points in the climate system.

– Apply some econometric methods to a different, but rather well-known, past climate change for which not only results from climate models are available but also various climate and environmental proxy data, and discuss the plausibility or implausibility of the results.

We investigate this latest option, choosing the Last Glacial Maximum (LGM, about 20,000 years ago) as a test past period. Indeed, on the one hand the global temperature increase projected by 2100 for unabated GHG emissions (scenario RCP8.5, Riahi et al., 2011) is roughly of the same amplitude, though of opposite sign, as the estimated temperature difference between the pre-industrial period and the LGM, that is about $4\,^\circ$C (Stocker et al., 2013). The magnitude of climatic and environmental changes during the last glacial-to-interglacial transition can thus provide an index of the magnitude of the changes that may occur for a warming of similar amplitude in 2100, as already postulated by Nolan et al. (2018). On the other hand, by design, statistical relationships linking climatic variables to economic damages could be applied either to a warming or to a cooling. Therefore, we try two of them for a hypothetical return to the LGM, except for the presence of northern ice sheets (see section 5), corresponding to a global *cooling* of $4\,^\circ$C in 2100.

---

[1] World GDP is probably a misnomer as we should rather mention Global World Product instead. We will nonetheless retain the common usage of 'World GDP'.

[2] The term damage function refers to the formal relation between climatic conditions and economic impacts, at the global level.

We chose two statistical functions linking GDP and temperature at the country level: the first one has been introduced by Burke et al. (2015) and formalizes the impact of temperature on country GDP growth; the second one by Newell et al. (2018) relates temperature to country GDP level[3]. The strength of this exercise lies in our ability to counter-check the results on potential damages under scrutiny with reconstructions from paleo-climatology.

The paper is structured as follows: section 2 briefly surveys the existing literature on climate change and economic damages, section 3 presents the methodology and data used here. The next section then describes the results obtained for our cooling scenario and section 5 compares these results with what is known of the Earth under such a climate situation. Section 6 then discusses our results in the light of the known strengths and weaknesses of such empirical functions while section 7 concludes.

## 2  Connecting climate change and economic damages

The literature on the broad topic of damage functions (see Tol (2018) for a review) can be organized into two broad approaches linking climate change to economic damages: an *enumerative* approach which estimates physical impact at a sectorial level, from natural sciences, gives them a price and then adds them up (e.g. Fankhauser, 1994; Nordhaus, 1994b; Tol, 2002), versus a *statistical* (or econometric) approach, based on observed variations of income across space or time to isolate the effect of climate on economies. (e.g. Nordhaus, 2006; Burke et al., 2015).

Each method has its pros and cons, some of which have already been acknowledged in the literature:

- The main advantage of the enumerative approach is to be based on natural sciences experiments, models and data (Tol, 2009). It distinguishes between the different economic sectors and explicitly accounts for climate impacts on each of them. Yet, results established for a small number of locations and for the recent past are usually extrapolated to the world and to a distant future in order to obtain global estimates of climate change impacts. The validity of such extrapolation remains dubious and it can lead to large errors. Moreover, accounting for potential future adaptations is a real challenge and therefore a major source of uncertainty in the projections. This method also implies to be able to correctly identify all the different channels through which climate affects the economy, which is by no means an easy task. And finally, it does not take into account interactions between sectors, nor price changes induced by changes in demand or supply (Tol, 2018).

- The statistical approach has the major advantage of relying on aggregates such as GDP per capita. There is no need to identify the different types of impacts for each economic sector and to estimate their specific costs. They rely on a limited number of climatic variables, such as temperature and precipitation, which are used as a proxy for the different climatic impacts. Adaptation is also implicitly taken into account, at least to the extent that it already occurred in the past. But as acknowledged by Tol (2018), one of the main weakness of some statistical approaches is that they use variations across *space* to infer climate impacts over *time*. This method also shares with the enumerative one the disadvantage of using

---

[3]Both Burke et al. (2015) and Newell et al. (2018) then create a world GDP value via a population-weighted average of the country GDP.

only data from the recent past, and hence from a period with a small climate change. The issue of future climatic impacts outside the calibration range of the function still holds.

Despite different underlying methodological choices, a large number of studies investigating future climatic damages conclude that global warming would cost only a few points of the world's income (Tol, 2018). A $3\,^{\circ}$C increase of the global average temperature in 2100 would allegedly lead to a decrease of the world's GDP by only 1-4%. Even a global temperature increase above $+5\,^{\circ}$C is claimed by certain authors to cost less than 7% of the world's future GDP (Nordhaus, 1994a; Roson and Van der Mensbrugghe, 2012).

Some statistical studies looking at GDP growth (e.g. Dell et al., 2012; Burke et al., 2015) emphasize the long-run consequences and lead to higher damage projections than those aforementioned. In particular, Burke et al. (2015) (hereafter BHM) evaluated the impact of global warming on growth at the country scale, using temperature, precipitation and GDP data for 165 countries over 1960-2010. According to their benchmark model, the temperature increase induced by strong GHG emissions (scenario RCP8.5) would reduce average global income by roughly 23% in 2100. This relatively high figure, however, is a decrease in *potential* GDP, itself identified with the projected growth trajectory according to the Shared Socio-economic Pathway 5 (SSP5, high growth rate, Kriegler et al., 2017). As a result, under a global temperature increase of about $4\,^{\circ}$C, only 5% of countries would be poorer in 2100 with respect to today, and global GDP would still be higher than today. It must be noticed that these results strongly depend on the underlying baseline scenario: if a lower reference growth rate is assumed (SSP3), the percentage of countries absolutely poorer in 2100 rises to 43%.

Capturing the impact of warming on growth rather than on GDP level may appear more realistic. Indeed, it allows global warming to have permanent effects and also accounts for resource consumption to counter the impacts of warming, reducing investments in R&D and capital and hence economic growth (Pindyck, 2013). There is however no consensus on the matter. In a recent work, Newell et al. (2018) (hereafter NPS) evaluate the out-of-sample predictive accuracy of different econometric GDP-temperature relationships at the country level through cross-validation and conclude that their results favor models with non-linear effects on GDP level rather than growth, implying, for their statistically best fitted model, world GDP losses due to unmitigated warming of only 1-2% in 2100.

Studies on future climate change damages to the global economy usually do not pretend to account for *all* possible future impacts. This is obvious for the enumerative methods applied at the global scale: being exhaustive is not realistically feasible. But this is also true for statistical approaches. Nordhaus (2006) for instance gives three major caveats to his statistically-based projections of climate change damages: 1) the model is incomplete; 2) estimates do not incorporate any non-market impacts or abrupt climate change, especially on ecosystems; 3) the climate-economy equilibrium hypothesis used is highly simplified. Burke et al. (2015) also acknowledge that their econometric model only captures effects for which historical temperature has been a proxy. Yet, despite these major caveats, results from both approaches are widely cited as "climate change" damages estimates (Carleton and Hsiang, 2016; Hsiang et al., 2017), as if they were really accounting for the whole range of future impacts, and some are used to estimate the so-called social cost of carbon (Tol, 2018). The authors themselves do not always clearly distinguish "climate change" impacts, which in the strict sense of the term should be applied to exhaustive estimates, from the non-exhaustive impacts accounted for by the specific chosen proxy variable.

In our view, in addition to these common semantic confusions, at least two of the aforementioned caveats are highly problematic: 1) extrapolating relationships outside their calibration range (which concerns both the enumerative and the statistical methods), and, 2) known and unknown missing impacts for which the chosen predicting climatic variables are not good explanatory variables. The fact that the channels of damages are not explicit in the statistical approach is convenient but also rather concerning: we simply *cannot* know which impacts are missed, except for a few of them (e.g. sea level rise).

A global warming of $4°C$ at the end of the century would drive the global climatic system to a state that has never been experienced in the whole human history, with growing concerns on the potential non-linearities in the way the Earth system as a whole may evolve: ecosystems have tipping points (e.g. Hughes et al. (2017); Cox et al. (2004)); the ice loss from the Greenland and Antarctic ice sheets has already clearly accelerated since the middle of the 2000s (Bamber et al., 2018; Shepherd et al., 2018); the projected wet-bulb temperature rise in the tropics could reach levels that do not occur presently on Earth and which would simply be above the threshold for human survival (Im et al., 2017; Kang and Eltahir, 2018). Thus, the question is: to what extent are we missing the point when using aggregated statistical approaches to estimate future damages?

Moreover, one could argue that we are not so sure about what a $+4°C$ warmer planet would look like, since we lack any analogue from the recent past. Yet, we actually have an example of a climate change of similar magnitude, albeit of a different sign, the last glacial period. In this paper, we focus on two representative examples of the statistical approach, the BHM and NPS functions, and use the LGM climate to test their relevance to assess the damages expected from a large and rapid climate change.

The choice of these two functions was based on the following considerations:

- While there might be controversies regarding the paper of Burke et al. (2015) related to the model specifications, interpretation and statistical significance of the results or even the validity of the approach, the approach is well published in leading peer-reviewed journals. Their work has been widely cited in the literature and has been used to compute the social cost of carbon (e.g. Ricke et al., 2018). The authors also published several other papers based on similar methodologies (e.g. Hsiang, 2016; Burke et al., 2018; Diffenbaugh and Burke, 2019).

- The function of Newell et al. (2018) has not been published in a peer-reviewed journal [4], but we considered it anyway because: 1) it belongs to the family of damage functions assuming an impact of climate on GDP level rather than growth, leading to very small damages; 2) it is based on the same data and methodology than Burke et al. (2015), hence simplifying the exercise.

We decided to perform an *ad absurdum* demonstration of the strong limitations of such approaches, because we believe that it is a useful complementary contribution to a more mathematical/statistical critique, which is not our purpose here. As documented by DeCanio (2003) for older functional forms, the literature on damage functions has had tremendous political implication, and even found its way in IPCC reports. Therefore, we believe it is important to add new elements to the existing critics.

---

[4]The paper can be downloaded here: https://media.rff.org/archive/files/document/file/RFF%20WP-18-17-rev.pdf

We chose to focus on the statistical approach because it inherently includes the effect of both cooling and warming. This is not the case for enumerative approaches, which are primarily designed for a warming and could not be applied to a cooling. They may nonetheless lead to implausible results as well, especially at the global scale (see the aforementioned caveats), but illustrating the disconnect between their damage projections and climate sciences would have required a different approach than the one we use here (e.g. questioning their assumptions, sector by sector, or providing damage estimates for impacts not taken into account).

## 3  Material and Methods

In order to assess the economic damages of a hypothetical return to an ice age, we compute the evolution of average GDP per capita by country, with or without the corresponding global cooling, following the methodology described in BHM, using the replication data provided with their publication. Details are available therein. We differ from BHM in two ways:

- For the simplicity of the demonstration, we chose to consider only one functional form linking temperature to GDP from BHM and NPS: we use either the BHM formula with their main specification (temperature impacts GDP growth, pooled response, short-run effect), whose results are the most commented in their manuscript, or the preferred specification of NPS (temperature impacts GDP level, best model by K-fold validation, full details in Newell et al., 2018).

- Our climate change scenario corresponds to a global cooling of $4\,^{\circ}$C, based on LGM temperature reconstructions and assuming a linear temperature decrease, instead of the climate projections for the RCP8.5 scenario. Following BHM, who consider only temperature projections for the assessment of future damages, we do not use LGM precipitation reconstructions.

Criticism of potential mathematical, variable choices or data issues in BHM or NPS work is beyond the scope of this paper. Our aim is limited to using their respective base equations as they are to test their realism for a large climate change scenario. Following BHM, we also use the socio-economic scenario SSP5 as a benchmark of future GDP per capita growth per country. SSP5 is supposed to be consistent with the GHG emission scenario RCP8.5 but it does not include any climate change impact, even for high levels of warming. Therefore, we can still use it in our glacial scenario without inconsistency.

The base case of BHM links the population-weighted mean annual temperature to GDP growth at the country level. Their model uses the following functional form:

$$\Delta ln(\text{GDP}cap_{i,t}) = f(T_{i,t}) + g(P_{i,t}) + \mu_i + \nu_t + h_i(t) + \varepsilon_{i,t}, \tag{1}$$

where $\Delta ln(\text{GDP}cap_{i,t})$ denotes the first difference of the natural log of annual real GDP per capita, i.e. the per-period growth rate in income for year $t$ in country $i$, $f(T_{i,t})$ is a function of the mean annual temperature, $g(P_{i,t})$ a function of the mean annual precipitation, $\mu_i$ a country-specific constant parameter, $\nu_t$ a year fixed effect capturing abrupt global events, and $h_i(t)$ a country-specific function of time accounting for gradual changes driven by slowly changing factors. BHM control for precipitation in

equation 1, because changes in temperature and precipitation tend to be correlated. Rather surprisingly, their study does not show a statistically significant impact of annual mean precipitation on per capita GDP.

In their base case model, $f(T_{i,t})$ is defined as:

$$f(T_{i,t}) = \alpha_1 \times T_{i,t} + \alpha_2 \times T_{i,t}^2 \tag{2}$$

Based on historical data, they determined the coefficient values to be $\alpha_1 = 0.0127$ and $\alpha_2 = -0.0005$.

Future evolution of GDP per capita in country $i$ and year $t$ between 2010 and 2100 is then given by:

$$\text{GDP}cap_{i,t} = \text{GDP}cap_{i,t-1} \times (1 + \eta_{i,t} + \delta_{i,t}), \tag{3}$$

with $\eta_{i,t}$ the business as usual country growth rate without climate change, according to SSP5 (taking into account population changes), and $\delta_{i,t}$ the additional effect of temperature on growth when the mean annual temperature differs from the reference average over 1980-2010, $T_{i,ref}$:

$$\delta_{i,t} = \alpha_1 \times (T_{i,t} - T_{i,ref}) + \alpha_2 \times (T_{i,t}^2 - T_{i,ref}^2), \tag{4}$$

It should be noticed that BHM do not take into account precipitation changes in their projection of future GDP.

The income growth-temperature relationship is a concave function of $T_{i,t}$, with an optimum temperature around $13\,°C$ (Fig.1). Therefore, for a country with a reference mean annual temperature below this GDP per capita-maximizing value (e.g. Iceland), the annual growth rate increases (resp. decreases) when the mean temperature increases (resp. decreases). This relationship is reversed for countries with a reference temperature above the optimum value (e.g. Nigeria). Note that for countries already close to the optimum temperature (like France), a small temperature change will have a very limited impact

on per capita GDP growth, but any major temperature change of several degrees will move them away from this optimum and have a negative impact on per capita GDP growth.

The preferred model of NPS links the mean annual temperature to the per capita GDP level, based on the same historical sample as BHM, and excludes any precipitation component. It links GDP in country $i$ at year $t$ to a polynomial function of mean annual temperature:

$$ln(\text{GDPcap}_{i,t}) = \beta_1 \times T_{i,t} + \beta_2 \times T_{i,t}^2 + ... \tag{5}$$

Based on historical data, NPS determined the coefficient values to be $\beta_1 = 0.008141$ and $\beta_2 = -0.000314$.

Using this formula, the future GDP per capita with climate change for the $21^{st}$ century, $\text{GDP}cap_{i,t}$, is expressed as:

$$\text{GDP}cap_{i,t} = \text{GDP}cap_{i,t}^* \times \exp\left[\beta_1 \times (T_{i,t} - T_{i,ref}) + \beta_2(T_{i,t}^2 - T_{i,ref}^2)\right], \tag{6}$$

with $\text{GDP}cap_{i,t}^*$ being the GDP per capita of the country without climate change, according to SSP5:

$$\text{GDPcap}_{i,t}^* = \text{GDPcap}_{i,t-1}^* \times (1 + \eta_{i,t}) \tag{7}$$

The NPS GDP-temperature relationships is also a concave functions of $T_{i,t}$, with an optimum temperature around $13\,°C$ (Fig.1). The shape is therefore similar to BHM, but the function is conceptually different since the impact of temperature is on

the GDP level instead of its growth rate. The SSP5 growth rate $\eta_{i,t}$ remains unaffected by climatic conditions and any negative temperature impact on year $t$ has no impact on the GDP per capita level at year $t+1$, which depends only on the underlying SSP5 scenario and on the temperature at year $t+1$.

To build our "glacial" scenario, we assume a linear decrease in temperature between 2010 (the end of the reference period) and the glacial state projected for 2100. For any year $t > 2010$, the country-specific mean temperature is therefore computed as:

$$T_{i,t} = \Delta T_i \times \frac{t - 2010}{2100 - 2010} + T_{i,ref} \tag{8}$$

with $\Delta T_i$ the population-weighted temperature anomaly of country $i$ at the LGM computed from Annan and Hargreaves (2013) (Fig.2) .

Similarly to Burke et al. (2015) who cap $T_{i,t}$ at $30\,^{\circ}\mathrm{C}$, the upper bound of the annual average temperature observed in their sample period, to avoid out of sample extrapolation, we cap the minimum possible value of $T_{i,t}$ at the lower bound of observations $(-5\,^{\circ}\mathrm{C})$.

## 4   GDP projections

All results are expressed as changes of average potential GDP per capita, based on the baseline SSP5 scenario which assumes no climate change. The impact on global average GDP per capita is a population-weighted average of country-level impacts.

Using the NPS specification, 34% of the countries see a lower income per capita than it would be without glacial climate change, but no country is poorer than today. The strongest impacts on GDP are projected on Northern countries: Canada and Norway for instance exhibit a potential GDP loss of about 8% in 2100. But at the global scale, the GDP loss projected in Northern countries is more than compensated by 1-2% GDP gain in most Southern countries (Fig.4(a)). All in all, the impact of the temperature decrease on the world's potential GDP is very limited, only about -1.8% in 2100 (Fig.3).

With BHM specification, projected impacts are much more severe in Northern countries: in the United States, Canada, Russia, and most of Europe GDP decreases range from 80% to nearly 100%, i.e. the impact of temperature on potential GDP growth is so large that it leads to a complete economic collapse (Fig.4(b) and Fig5). Similarly, stronger positive effects are projected in Southern countries, with large GDP increases for most of them in 2100 (Fig.4(b)): e.g. +254% in Gabon, +314% in Ghana, +267% in India, +300% in Laos, +366% in Mali or +400% in Thailand. China is the sole country where potential GDP remains roughly unchanged with impacts smaller than 1%. Globally 31% countries exhibit lower income per capita than projected without climate change and 17% are poorer than today. Losses in Northern countries drive a decrease in the world's GDP during the first half of the century, with maximal global damages around 2050, at about -4%. In the second half, however, positive impacts in Southern countries more than over-compensate damages in the North and as a consequence average potential GDP per capita gains +36% in 2100 at the world level with respect to the baseline scenario (Fig.3).

## 5 Comparison with LGM conditions

To assess the credibility of these results we now survey the environmental conditions that human beings would have to face on our planet under our theoretical scenario, taking what is currently known of the LGM as a reference and considering both climate and ecosystem changes. Ecosystem changes where then driven by both climate change and the impacts of low atmospheric $CO_2$ concentrations on photosynthetic rates and plant water-use efficiency (Jolly and Haxeltine, 1997; Cowling and Sykes, 1999; Harrison and Prentice, 2003; Woillez et al., 2011) but, in order to simplify our argument, we do not distinguish between these two effects in our description of a world cooled by $4\,°C$.

Many reconstructions of the climatic and environmental conditions at that time are available (Kucera et al., 2005; Bartlein et al., 2011; Prentice et al., 2011; Nolan et al., 2018; Clark et al., 2009), as well as numerous modeling exercises (Braconnot et al., 2007; Kageyama et al., 2013; Annan and Hargreaves, 2013; Kageyama et al., 2018). Despite remaining uncertainties and discrepancies, data-based reconstruction and modeling results provide a fairly good picture of the Earth during the LGM.

The most striking feature of the last glacial world was the existence of large and thick ice sheets in the northern hemisphere (Peltier, 2004; Clark et al., 2009). Of course, reaching the full extent of the LGM ice sheets, which depends on both static snow accumulation and ice viscous spreading, would require tens of thousands of years, not a century. Therefore, as a simplification for the sake of the demonstration, we assume we have reached the LGM climate equilibrium, except for the ice-sheet thickness and extent and associated sea-level drop, since the timing is obviously too short. Such a simplification implies some inconsistency, since the LGM climate also depends on the albedo and elevation feedbacks from the ice sheets. We also acknowledge that 1) the response of the Earth system to a forcing that would lead to a $4\,°C$ cooling in 2100 would be different from the LGM, depending on the type of forcing, and therefore the LGM is not a perfect reverse analog of a future at $+4\,°C$; 2) it took much more than a century to move from the LGM to the Holocene, our current interglacial period. However the projected rate of global warming for the RCP8.5 scenario is actually faster than any glacial-inter glacial changes that occurred naturally during the last 800,000 years: about 65 times as fast as the average warming during the last deglaciation (Nolan et al., 2018). Besides, the level of warming in 2100 for the RCP8.5 scenario might exceed $+4\,°C$, especially if strong positive feedback loops lead to the crossing of planetary thresholds hence driving Earth in a "hothouse" state (Steffen et al., 2018; Schneider et al., 2019). Accordingly, using the LGM-to-present environmental changes as an index of future changes might even be considered as conservative.

With these caveats in mind, let us now take a closer look at the most obvious consequences of our scenario for human societies.

Results from climate models show that the surface mass balance of the northern LGM ice sheets was positive over most parts, outside the ablation zones on the edges, with annual accumulation rates of water equivalent of a few tens of cm per year (e.g. Calov et al., 2005). Therefore, in our thought experiment, snow accumulation on regions corresponding to the LGM ice-sheet extent would reach a few meters at the end of the century. Moreover, in the most central regions, the decrease of the mean annual temperature would be greater than $20\,°C$ (Fig.2) and the cold would be hard to cope with. We presume that regions experiencing either such low temperatures and/or being buried under a thick permanent and growing layer of snow would

become rather unsuitable for most modern economic activities. The impacted regions would be: Canada, Alaska and the Great Lakes region of the United States, the states north of $40\,°N$ on the East coast, the Scandinavian countries, the northern part of Ireland and of the British islands, half of Denmark, the northern parts of Poland and the north-east territories of Germany, all of the Baltic countries as well as the north-eastern part of Russia. We assume that alpine regions that were widely covered by glaciers during the LGM, i.e. Switzerland and half of Austria, would in our scenario also experience several meters of snow accumulation. All these regions would become unsuitable for most of the millions of people who currently live there, and access to their present natural resources would be very difficult, if not impossible. By comparison, nowadays regions with a mean annual temperature below about $5\,°C$ have a very low population density (Xu et al., 2020).

Shipping routes in the North Atlantic would also be disrupted by the southern expansion of sea-ice up to $50\,°N$ in winter (Gersonde and De Vernal, 2013) and calving icebergs.

In Europe, the mean annual temperature would decrease by $4-8\,°C$ in the Mediterranean region, by $8-12\,°C$ over the western, central and eastern regions and by more than $12\,°C$ over northern countries (Fig.2). For France for instance, whose current mean annual temperature is about $11\,°C$, the temperature decrease would thus correspond to a shift to the current mean temperature of northern Finland. Over western Europe, the mean temperature of the coldest month would decrease by $10-20\,°C$ (Ramstein et al., 2007) and the mean annual precipitation would decrease by about 300 mm/year (Wu et al., 2007). Forests would be highly fragmented, replaced by steppe or tundra vegetation (Prentice et al., 2011). The southern limit of the permafrost would approximately reach $45\,°N$, i.e. the latitude of Bordeaux (Vandenberghe et al., 2014). In such a context, maintaining European agriculture, at its present state, among other human activities, would be a costly and technically highly demanding challenge. Energy needs for heating would tremendously increase, current infrastructures would be damaged by severe frost and there is no doubt that Europe would no longer sustain its current population on lands preserved from permanent snow accumulation.

In Asia, similar problems would occur, with a decrease in mean annual temperature between 4 to $8\,°C$ over most Chinese regions for instance (Fig.2). The boreal forest would progressively vanish, replaced by steppe and tundra (Prentice et al., 2011). Permafrost would extend in the North-East and North China, up to Beijing, as well as in the west of the Sichuan (Zhao et al., 2014). As a result, rice cultivation in the northern province of Heilongjiang for instance, currently above $20.10^6$ tons/year, i.e. about 10% of the national production (Clauss et al., 2016), would no longer be possible, among other crops. Permafrost would not stretch out to the whole densely populated North China plains, but the cold and dry climate there would nonetheless prevent rice cultivation. The discharge of the Yangtze River at Nanjing would be less than half its present-day value (Cao et al., 2010), questioning current hydroelectricity production. In short, current livelihoods in these regions would no longer be sustainable and population would probably be much lower than today.

Temperature changes in the tropics would be rather moderate, with a cooling of $2.5-3\,°C$ (Wu et al., 2007; Annan and Hargreaves, 2013) (Fig.2). This temperature decrease might be considered as good news, and is indeed the driver of the GDP increase simulated in tropical countries with both specifications we considered (Fig.4). However, tropical temperature decrease would come with strong changes in the hydrological cycle, casting some doubts on such an optimistic view. The inter-annual rainfall variability in East Africa would be reduced (Wolff et al., 2011), but so would be the mean rate; the Southwest Indian

monsoon system would be significantly weaker over both Africa and India (Overpeck et al., 1996); the Sahara desert and Namib desert would both expand (Ray and Adams, 2001); annual rainfall over the Amazon basin would strongly decrease (Cook and Vizy, 2006). Compared to their modern extension, the African humid forest area might be reduced by as much as 74%, and the Amazon forest by 54% (Anhuf et al., 2006).

Globally, the planet would appear considerably more arid (Kageyama et al., 2013; Ray and Adams, 2001; Bartlein et al., 2011). However, this widespread increase in aridity is debated since the reduction of the atmospheric demand for evaporation because of lower temperatures could compensate the precipitation decrease and drier places in terms of precipitation at the LGM were not always drier in terms of hydrology (Scheff et al., 2017; McGee, 2020). As already mentioned previously, vegetation changes are also driven by the decrease in atmospheric $CO_2$, which can bias aridity increase inferred from pollen proxy data. Yet, many places were indeed hydrologically drier at the LGM, including some currently densely populated areas. The southward spread of the extra-arid zone of the Sahara desert for instance is estimated to 300-450 km (Lioubimtseva et al., 1998). For India, LGM data are rather sparse. Simulations results suggest that there was indeed a large decrease in runoff across monsoonal Asia (Li and Morrill, 2013), in agreement with marine proxy data from the Bay of Bengal suggesting a reduction in fluvial discharge (Duplessy, 1982). This could mean less flooding during the monsoon season, but also a decrease of water resources during the dry season, in areas where droughts are already an issue nowadays. In regions already dry today, like Pakistan or north-western India, it seems that the last glacial conditions were even more arid (Ansari and Vink, 2007). In Indonesia, simulations results show a decrease in the precipitation minus evaporation (Scheff et al., 2017) but vegetation changes depends on the location (Dubois et al., 2014), and we cannot really inferred potential impacts for human populations.

The planet would also appear much dustier than today, with probably more frequent and/or intense dust storms which would impact soil erosion rates, health, transportation and electricity generation and distribution (Middleton, 2017). Potential increase of dust source areas include north-east Brazil, central and southern South America, southern Africa, central Asia and the Middle East, Australia, China and south-eastern Asia (Harrison et al., 2001).

Thus, overall, postulating that cooling would drive a large GDP surge in tropical countries, as simulated with BHM specification, is highly questionable.

## 6 Discussion

In summary, our hypothetical ice-age scenario corresponds to strong and widespread changes in climatic conditions, not only in temperature, driving major environmental changes (Nolan et al., 2018). In such conditions, neither the results obtained with the BHM and NPS functions nor the baseline GDP scenario (SSP5) appear as plausible projections.

### 6.1 Temperature-GDP level relationship

We argue that the disruptions in the living conditions on our planet, as briefly described above, cannot plausibly result in a small decrease of less than 2% in the world potential GDP per capita in 2100, as inferred from the NPS specification. According to these results, Canada would experience only a 8% decrease in its potential GDP per capita, despite its infrastructure being

buried under snow, its natural resources being inaccessible or disappeared and tremendous frost. Such estimations of climate damages remain utterly unrealistic even if we were ready to consider optimistic adaptation skills of human societies that would prevent them from social calamities such as revolutions, famines or wars. Our results illustrate how the idea that climate influences only the level of economic output and has no impact on economic growth trajectory is not appropriate for a large climate change. The complete failure of this approach to provide plausible results for a cooling discredits its reliability to account for the impact of a global warming of similar magnitude, which would without doubts drive environmental changes as huge as the one we listed above for the LGM.

## 6.2 Temperature-GDP growth relationship

The BHM specification gives somewhat more plausible results for Northern countries, with the projection of a complete collapse of their economies, in agreement with the prospect of permanent snow accumulation, very low temperatures and large ecosystem shifts. However, we have serious doubts on the (very) large GDP per capita increase predicted in tropical countries, given the strong decrease in precipitations in many places and global desert expansion, threatening in particular water resources and agriculture. How can we reconcile, for instance, the projection of a GDP increase of more than 300% in sahelian countries with a southward expansion of the Sahara desert of about 400 km? The BHM set-up focuses on damages driven by temperature change only (or changes for which temperature is a proxy) and does not take into account precipitation changes for climate change projections. The author's study did not find that mean annual precipitations had a significant effect on the economy in the last decades, a result rather surprising, considering the strong impacts that droughts or extreme precipitations may have. This could be due to the fact that the mean precipitation at the country scale is not an appropriate variable, since it does not necessarily capture correctly seasonality changes or extreme events for instance. In any case, should precipitation effects be negligible for the recent past, they cannot be ignored in the case of major hydrological changes that would also drive radical ecological shifts.

Similarly, the absence of damages in China can hardly be conceptually reconciled with both deserts and permafrost expansion, which should very probably have strong negative impacts on agriculture in the north and north-east of the country, or the strong decrease in fluvial discharge.

Moreover, the complete collapse of (at least) the northern nations, including expected massive migrations of millions of people outside these regions, would be expected to have serious economic and geopolitical consequences at the global scale, that we can hardly imagine being very positive. The statistical method of BHM capture the present-day political and economic relationships between countries, but it cannot account for future changes in these relationships, a major deficiency in a globalized world.

It is difficult to imagine how the world could be globally much wealthier than it would have been without such disruptions in climatic and ecological conditions, especially if most places are no longer suitable for agriculture, as it may have been the case during the Pleistocene (Richerson et al., 2001). Agriculture may account for only a few percentages of GDP in present-day developed countries, but food production is obviously the first need of any society. We therefore conclude that, despite its endeavor toward realism, the BHM function does not provide results more convincing than the NPS one.

### 6.3 Cooling vs warming: symmetry or asymmetry of impacts ?

From a physical point of view, there is no *a priori* reason to postulate that a global warming and cooling of similar magnitude would have similar huge impacts. However, the symmetry is implicitly assumed by the GDP-temperature relationship itself: it was built on both negative and positive temperature anomalies and therefore, by design, it cannot be assumed that such a function could provide relevant damage estimates for a warming, but not for a cooling (or the other way around). Moreover, when considering some of the most dramatic climate projections for the RCP8.5 scenario at the end of the century, it seems rather plausible that such a warming would have similar strong impacts as a cooling of similar magnitude:

- On the one hand, as discussed in section 5, for our LGM scenario large parts of North America and Europe would become rather unsuitable for a large human population and most modern economic activities. Currently 37 millions of people live in Canada and about 30 millions in northern Europe, where we can reasonably assume that only a small population could remain. Maintaining a total population of more than 700 millions of people in Europe despite the extremely cold temperatures in winter is doubtful, even if the number of people that could still live there (probably mostly in southern Europe) remains speculative. Similarly, in India, data suggest that the north-western part of the subcontinent experienced extreme desert conditions (Ansari and Vink, 2007), which would have strong negative impacts for its current 70 millions of inhabitants (state of Rajasthan).

- On the other hand, for global warming, the number of people currently living in areas which may be exposed to permanent inundation for a sea level rise of 1.46 m in 2100 has been estimated to 340 millions (Kulp and Strauss, 2019), a number that would be further increased for higher sea level rise values (Bamber et al., 2019). But the most alarming projections are maybe the ones concerning future heat stress: according to Mora et al. (2017), temperature and humidity conditions above potentially deadly threshold for humans could occur nearly year-round in humid tropical areas, including some of the most densely populated areas. How people could adapt to such unprecedented climatic conditions remains an open question.

### 6.4 General issues

Whether temperature changes impact the GDP *growth* or *level* is actually a debate of little relevance. In both cases, the use of the mean temperature at the country scale as a proxy for climate effects turns out to provide a highly insufficient picture in the case of a large climate change and leads to a large underestimation of the risks to lives and livelihoods.

This failure can be attributed to different issues of such statistical approaches:

- To our knowledge, there is currently no publications on potential issues in the statistical model itself. To this respect, comments made by one anonymous referee regarding stationarity and the use of control variables (see the public discussion of this paper as well as section B.2 of the supplementary information of Burke et al. (2015)) seem interesting and worth investigating. As these considerations fall outside the scope of this paper, we leave them for further research. Beyond any purely mathematical consideration, it is interesting to note that both BHM and NPS considered only the

mean annual temperature and precipitations to build their respective model. Yet, we speculate that these are not good proxies for climatic variables that would have strong economic impacts, such as seasonality, extreme precipitation events, droughts or heatwaves.

– As mentioned in section 2, one of the serious limitation of these statistical approaches is that they rely on climatic variations over space to extrapolate over time. Indeed, BHM argue that, for most countries in their sample, a global warming of $4\,°C$ takes them out of their own historical range of temperature, but that they still remain within the worldwide distribution of historical temperatures. For that reason, they consider that there is no extrapolation out of sample for these countries. If a country gets warmer, the economic impacts can be deduced, they assumed, from past observations in another country whose past temperature was similar. Only a few hottest countries would reach temperature outside the worldwide historical range, and for this category they chose not to extrapolate but to cap future temperature at the upper bound observed in the sample period. As already pointed out by Pezzey (2019), such assumption is actually untestable. One could argue that human adaptation capacities would succeed in maintaining climate-economy equilibrium even in a changing climate. This hypothesis would be very doubtful in the case of long-living infrastructures facing rapid climate change and certainly does not hold for ecosystems, one of the channels through which climate change impacts the economy. Ecosystems simply cannot adapt quickly enough to a climate change as fast as $+4\,°C$ in a century. The speed of forest migration for instance is a few hundreds of meters per year (e.g. Brewer et al., 2002) while temperature change in 2100 according to the RCP8.5 scenario would correspond to a displacement of more than 1000 km of current temperature zones. The ecosystem-climate equilibrium is not valid on the timescale of a century and therefore, we argue that this issue is in itself sufficient for the extrapolation from space to time to be unwarranted.

– BHM and NPS functions are based on economic data from societies adapted to their current environment. The alleged statistical relationship between GDP per capita and temperature is established for stable ecological conditions and is therefore hardly relevant to assess damages on societies who will experience decades of drastic changing climate and ecosystems and having to re-adapt endlessly to ephemeral new living conditions. It should also be stressed that, as illustrated in Burke et al. (2015), the results obtained with their methodology strongly depends on the assumed reference GDP growth rate without climate change. There are evidences that economic growth rates are path-dependent (Bellaïche, 2010), therefore in this case it makes no sense to apply a correction to a baseline growth rate which remains unaffected by the damages that occurred the previous years.

– What econometrics maybe show is that, *for the few last decades*, with a still relatively stable climate, interannual weather variability does not have a strong impact on the economy. We cannot extrapolate from this limited sample what would be the consequences of new climatic conditions completely out of sample and never experienced by humans so far, such as those which could occur in the tropical regions for unabated GHG emissions (Mora et al., 2017; Im et al., 2017; Kang and Eltahir, 2018).

– Beyond the issue of the relevance of the use of mean temperature and precipitation changes as proxies for other climatic variables, there are obviously impacts not accounted for (DeFries et al., 2019) such as glacier melting and resulting water challenges once they would have vanished, potential tipping points (Lenton et al., 2008; Steffen et al., 2018) such as a rapid melting of the Greenland or Antarctic ice sheet that would trigger fast sea level rise (Sweet et al., 2017), thawing permafrost, stronger tropical cyclones, ocean acidification and ecosystem shifts. It is therefore very misleading to consider that they allow to quantify "climate change" economic damages. At best, they might give an insight of damages for which temperature has been historically a proxy, and this is highly insufficient, conveying a false picture of the potential risks.

## 7 Conclusions

Should GHG emissions continue unabated, the climate change expected for the end of the century will be of similar magnitude than the last deglaciation, which did not occurred in a century but in about 10,000 years. Such a rapid change has no equivalent in the recent past of our planet, even less so in human history. Trying to establish a robust assessment of future economic damages based on aggregate statistics of a few decades of GDP and climate data, as attempted by econometric approaches, is probably doomed to failure, even more so when considering only mean annual temperature as a proxy for climate change. Such methodologies seem irrelevant for what lies ahead, since they fail to account for the largest potential impacts of climate change, as was already pointed out by DeFries et al. (2019). In order to strengthen this point, we have used an *ad absurdum* example of a hypothetical return to the climatic and environmental conditions of the LGM, except for the presence of the northern ice sheets and associated sea-level drop, corresponding to a global cooling at speed and magnitude equivalent to what the business-as-usual scenario of the IPCC announces. The comparison between the results obtained with two different statistical temperature-GDP relationships for our scenario and what we know of the Earth during the LGM suggests that both approaches are severely underestimating the impact of climate change. We can therefore conclude that temperature only is a very bad proxy to estimate damages of a major climate change at a country scale or at the global scale and should not be used for that purpose. More generally, several issues inherent to statistical approaches cast strong doubts on potential significant improvement. In this context, empirically estimating the aggregated relationship between economic activity and weather variables to project future damages is at best useless, at least from a policy point of view. Economists should hence refrain from using existing statistical damage functions to infer the global impacts of climate change or to compute optimal policy.

To summarize, our work has proven by absurdum the strong limitations of statistically-based methods to assess quantitatively future economic damages. In our view, a more modest and realistic ambition could be endorsed by integrated assessment scenarios, namely that of making an educated guess on the lower-bound of such damages at regional, rather than global, scales where the uncertainty surrounding prospective estimations may be more easily dealt with. This alternate kind of approach would be closer to the enumerative one mentioned in the introduction. This ideal approach should however not merely use sectorial statistical relationships established for the recent past, as is often currently done. They would otherwise underestimate damages just as aggregated statistical method do. Instead, they should account for tipping points or potential cascading effects

and should definitely be consistent with the future described by climate and ecological sciences (Stocker et al., 2013). But as already pointed out by Pezzey (2019) it is highly probable that high levels of uncertainties will remain and some risks *"are currently impossible to assess numerically, which economists need to acknowledge with greater openness and clarity"* (DeFries et al., 2019).

*Author contributions.* Marie-Noëlle Woillez designed the study, performed the simulations and wrote the manuscript, Gaël Giraud and
475 Antoine Godin participated in the discussion on the paper content and in the writing of the manuscript.

*Competing interests.* The authors declare that they have no conflict of interest.

*Acknowledgements.* We thank Mikhail Verbitsky and two anonymous reviewers for their comments to improve this paper. We also thank Antonin Pottier for useful comments and discussion on the initial version of this manuscript.

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

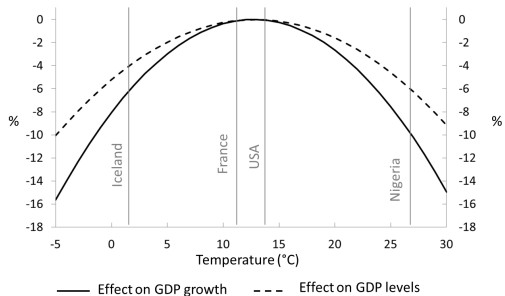

**Figure 1.** GDP per capita-temperature relationships, growth (BHM) and level (NPS) effects (percentage points). The curves are shown on the same plot but are not directly comparable, since their respective impact on GDP is fundamentally different. Vertical lines indicate average temperature for 4 selected countries. Each curve has been normalized relative to its own peak.

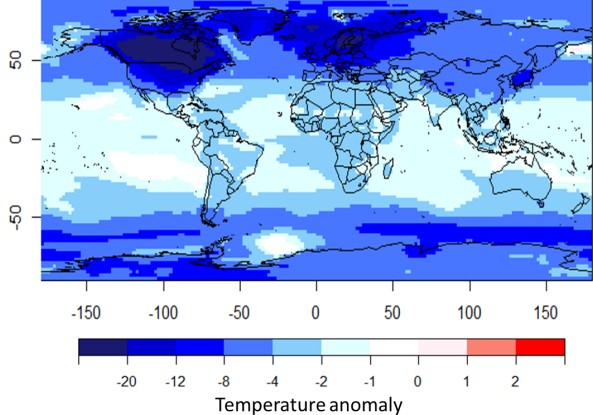

**Figure 2.** Reconstruction of the Last Glacial Maximum surface air temperature anomalies (°C) based on multi-model regression. Data source: Annan and Hargreaves (2013).

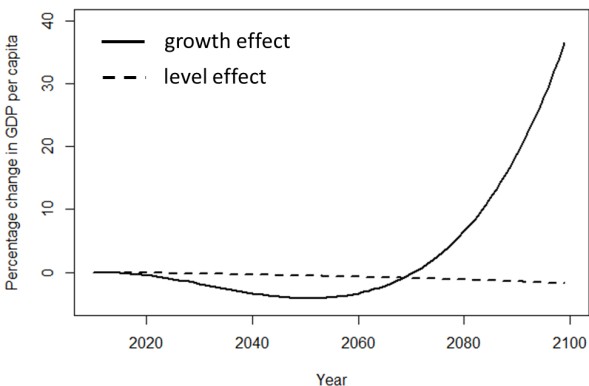

**Figure 3.** Percentage change in average GDP per capita (world level) for a global cooling of $-4\,°\mathrm{C}$ in 2100 as projected from non-linear effects of temperature on GDP level (dashed line, Newell et al. (2018) specification) or growth (plain line, Burke et al. (2015) specification). Reference GDP path according to the SSP5 scenario.

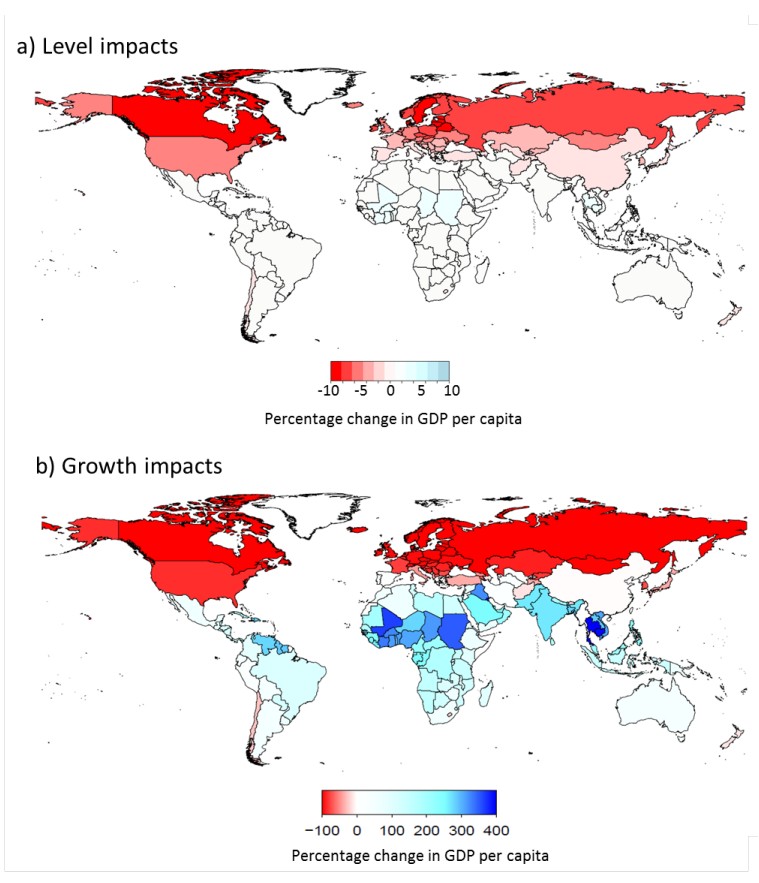

**Figure 4.** Projected impacts of a $-4\,^{\circ}\mathrm{C}$ global cooling on GDP per capita in 2100. Changes are relative to projections without climate change according to SSP5. a) Changes according to NPS specification (GDP level effects); b) changes according to BHM specification (GDP growth effects). NB: color scales have different maximum and minimum values for easier visualization.

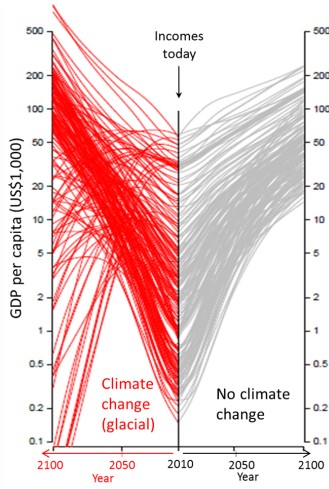

**Figure 5.** Country-level average income projections with and without temperature effects of a "glacial" climate change. Projections to 2100 according to SSP5 scenario, assuming high baseline growth and fast income convergence. Centre is 2010, each line is a projection of national income. Right (grey) are incomes under baseline SSP5 assumptions, left (red) are incomes accounting for a non-linear effects of projected cooling on GDP growth.