# Peer review of "Economic impacts of a glacial period: a thought experiment to assess the disconnect between econometrics and climate sciences"

_Earth System Dynamics, 2020_

## Short Comment (SC1) · 29 May 2020

The authors endeavor to study the limitations of some quantitative methods of assessing future economic damages using "an ad absurdum example of a hypothetical cooling of climate at speed and magnitude equivalent to what the business-as-usual scenario of the IPCC announces."

Though ad absurdum examples are always entertaining and occasionally useful, the "degree" of absurdity should be constrained by the laws of physics. From this standpoint, fast cooling of the planet toward the end of this century, though hard to imagine, is not precluded by physics. The real absurdity, which is not supported by physics

and therefore invalidates the study, appears elsewhere. The authors suggest that "...
the regions covered by ice at the LGM would in our scenario be buried under several
meters of snow at the end of the century...The impacted regions would be: Canada,
Alaska and the Great Lakes region of the United States, the states north of 40N on
the East coast, the Scandinavian countries, the northern part of Ireland and of the
British islands, half of Denmark, the northern parts of Poland and the north-east terri-
tories of Germany, all of the Baltic countries as well as the north-eastern part of Russia,
Switzerland and half of Austria". It appears that the authors think that an ice age begins
immediately and simultaneously on ∼50 mln square kilometers when the winter snow
is not completely melted during the summer and over time becomes what we know as
Laurentide and Scandinavian ice sheets. This is not how ice-age physics works. The
timing of the ice ages is defined by the speed of the moving horizontal boundary of
the spreading viscous ice media, not by the snow growth from the ground. To spread
all over the areas mentioned above, would take about 100,000 years, not 1,000 years
as the authors suggest. Therefore, though the authors call this vast permanent snow
coverage the "most obvious consequences for human societies", it is in fact far from
obvious and all of these regions may be permanent-snow-free for a very long period of
time. Even the emergence of smaller, nucleus, glaciers (that do grow from the ground)
is not granted because the cooling may reduce snow precipitation rates in polar re-
gions instead of increasing them. In short, the climate system is non-linear; the ice
ages begin when the global temperature is high and end when it is low. Anthropogenic
global warming, if extended, may preclude next ice age; it doesn't necessarily mean
that anthropogenic global cooling would "instantaneously" generate one.

I understand that this paper is not about ice-age physics, and the authors want to
make a (probably valid) point about the inconsistency of some economic models, but
their choice of the thought experiment is very unfortunate. As economists, they want to
"...conclude that temperature only is a very bad proxy to estimate damages of a major
climate change at a country scale or at the global scale and should not be used for that
purpose" but, as climatologists, they make exactly the same mistake, assuming that

temperature only (-4 C) would bring our climate exactly where it was 20,000 years ago.

---

## Author Comment (AC1) · 3 Jun 2020

***The comments of Mikhail Verbitsky are reproduced below in black. Our responses are in blue.***

The authors endeavor to study the limitations of some quantitative methods of assessing future economic damages using "an ad absurdum example of a hypothetical cooling of climate at speed and magnitude equivalent to what the business-as-usual scenario of the IPCC announces."

Though ad absurdum examples are always entertaining and occasionally useful, the "degree" of absurdity should be constrained by the laws of physics. From this standpoint, fast cooling of the planet toward the end of this century, though hard to imagine, is not precluded by physics. The real absurdity, which is not supported by physics and therefore invalidates the study, appears elsewhere. The authors suggest that "…the regions covered by ice at the LGM would in our scenario be buried under several meters of snow at the end of the century…The impacted regions would be: Canada, Alaska and the Great Lakes region of the United States, the states north of 40N on the East coast, the Scandinavian countries, the northern part of Ireland and of the British islands, half of Denmark, the northern parts of Poland and the north-east territories of Germany, all of the Baltic countries as well as the north-eastern part of Russia, Switzerland and half of Austria". It appears that the authors think that an ice age begins immediately and simultaneously on 50 mln square kilometers when the winter snow is not completely melted during the summer and over time becomes what we know as Laurentide and Scandinavian ice sheets. This is not how ice-age physics works. The timing of the ice ages is defined by the speed of the moving horizontal boundary of the spreading viscous ice media, not by the snow growth from the ground.To spread all over the areas mentioned above, would take about 100,000 years, not 1,000 years as the authors suggest. Therefore, though the authors call this vast permanent snow coverage the "most obvious consequences for human societies", it is in fact far from obvious and all of these regions may be permanent-snow-free for a very long period of time.

Even the emergence of smaller, nucleus, glaciers (that do grow from the ground) is not granted because the cooling may reduce snow precipitation rates in polar regions instead of increasing them. In short, the climate system is non-linear; the ice ages begin when the global temperature is high and end when it is low.

Thank you for your comment; we agree that ice sheets do not grow only because of static snow accumulation and the role of the viscous spreading should indeed be mentioned in the text. We also perfectly acknowledge that reaching the last glacial maximum northern ice sheets geometry would not be possible within 100 years, as stated L. 214 of our paper. By "millennia" we meant "thousands of years", the text should be modified for greater clarity. L.214 would then be modified as follows (in italics): "Of course, the growing and *spreading* of large ice sheets actually requires *thousands of years*, not a century."

We also agree that ice sheets do not build quickly as soon as snow accumulates on the ground. However, according to the Milankovitch theory, ice ages are triggered by the reduction of summer insolation at high latitudes, which allows winter snow to persist in summer and then eventually, because of various positive feedbacks (e.g. Khodri et al., 2001; Calov et al., 2005), to the ice-sheet build-up and full glaciation. Snow accumulation is not sufficient to build an ice sheet, but it is necessary.
Indeed, the last ice age was drier than present-day climate (e.g. Kageyama et al., 2020) and precipitation decrease could prevent snow accumulation on the areas listed in our paper.

However the LGM simulations of Calov et al. (2005) show that the area of positive annual mass balance correspond to the area of the ice sheets (except on the edges of the Fennoscandian ice sheet). In another context, Robock et al. (2009) also simulated the persistence of snow in the midlatitudes of the North hemisphere in response to a massive volcanic eruption, because of very low temperatures. Therefore, our assumption does not seem utterly unreasonable, even if it is of course a rough first-order estimate.

Moreover, the direct comparison between the timing of the last glacial cycle and our thought experiment is not possible since the slow ice-sheet build-up during the last glacial period was triggered by the slow evolution of the orbital forcing. Their evolution was far from linear during the ~100 ky between the last interglacial and the LGM: proxy data show a fast growth of the ice sheets between 120-110 ky, with a 30-60 m of sea level drop. Simulations of the glacial inception by Calov et al. (2005) show that "*Between 118 and 117 kyr BP the land area covered by ice increases by more than $4.10^6$ $km^2$ in just a few hundred years, reaching approximately 30% of the area of LGM ice sheets in North America*". The temperature decrease during the LGM compared to present-day would be a much stronger driver than the orbital forcing at the last glacial inception, and the ice-sheet growing and spreading could then be much faster than what was observed for the last glacial inception. However, to our knowledge, no modelling study is currently available in the literature to test this hypothesis. The closest comparison could be with the strong cooling induced by a massive volcanic eruption (Robock et al., 2009) or a nuclear war (Robock et al., 2007), but in such cases the pattern of temperature and precipitation changes is different from the LGM and lasts only a few years.

In any case, the estimated -4°C of global cooling during the LGM corresponds to a climate more or less at equilibrium, where all feedbacks had enough time to act (snow and ice albedo , decrease of greenhouse gases, vegetation changes, elevation of the ice sheets…etc). In our work, we chose to assume equilibrium as a simplification for the sake of the demonstration and to have a known period to estimate the plausibility of econometrics projections. The only exception to this assumption are the ice sheets and associated sea level drop, as the timing is obviously too short. The LGM ice-sheet extension is used as a rough constraint for the location of the areas where snow accumulation would occur, not taking into account the role of ice dynamics during the LGM, which should be mentioned in the text for more clarity. But we cannot constrain the hypothetical rate of snow accumulation, (taking into account a drier climate) without ad hoc experiments with a climate model, which is beyond the scope of the paper. In any case, even if some of the regions listed in our paper would remain snow-free for a long time, the cold would anyway be a strong constraint to economic activities, even if not as dramatic as with a thick permanent snow cover. By comparison, currently most of the high northern latitudes have a very low population density (see also the recent work by Xu et al. 2020 on human ecological niche).

We suggest adding at L.227 the following sentence (in italics): "We presume that the regions covered by ice at the LGM would in our scenario be buried under several meters of snow at the end of the century. *This is a rough first-order estimate, not taking into account the role of ice dynamics in the spreading of the ice-sheets over the north hemisphere during the last glacial period*".

Anthropogenic global warming, if extended, may preclude next ice age; it doesn't necessarily mean that anthropogenic global cooling would "instantaneously" generate one.
I understand that this paper is not about ice-age physics, and the authors want to make a (probably valid) point about the inconsistency of some economic models, but their choice of the thought experiment is very unfortunate. As economists, they want to ": : :conclude that temperature only is a very bad proxy to estimate damages of a major climate change at a country scale or at the global scale and should not be used for that

purpose" but, as climatologists, they make exactly the same mistake, assuming that temperature only (-4°C) would bring our climate exactly where it was 20,000 years ago.

We do not assume that such a cooling would be of anthropogenic origin. Actually, we do not propose any physical mechanism. We make the hypothesis of a return to the LGM, even if physically implausible, and merely have a look at what the consequences would be according to econometrics. Our main focus is on highlighting the unrealistic results obtained with statistical damage functions for a climatic change symmetrical to the RCP8.5 (when looking only at the mean temperature), not to discuss the physical mechanisms that could trigger such a change. Although not fully comparable, our approach is inspired from Nolan et al. (2018), who use the ecosystem changes during the last glacial-to-interglacial transition as a proxy to assess the risk of future major ecosystem transformations worldwide in the case of unabated greenhouse gas emissions.

However, our working hypothesis probably needs to be clarified in the introduction. We do not assume that temperature change would trigger a return to a glacial climate. Rather, we assume a return to the last glacial maximum, with all its consequences (including precipitation and ecosystem changes for instance), and use it as a benchmark to test econometric models based on mean annual temperature, to illustrate that looking at temperature changes only leads to unrealistic impacts on GDP. Therefore, we did not consider only temperature changes to quantitatively estimate the impacts, but also precipitation, vegetation, permafrost or desert area changes, since lower temperatures and large ice sheets are not the only features of the last ice age. This could be clarified in the text by modifying L.37 as follows: "*Therefore, we can try them for a hypothetical return to the LGM, corresponding to a cooling of 4°C in 2100*". In the conclusion, L.353 could also be modified as follows: "*In order to strengthen this point, we have used an ad absurdum example of a hypothetical return to the climatic and environmental conditions of the last glacial maximum, corresponding to a global cooling at speed and magnitude equivalent to what the business-as-usual scenario of the IPCC announces*".

Calov, R., Ganopolski, A., Claussen, M., Petoukhov, V., & Greve, R. (2005). Transient simulation of the last glacial inception. Part I: glacial inception as a bifurcation in the climate system. *Climate Dynamics*, *24*(6), 545-561.

Kageyama, M., Harrison, S. P., Kapsch, M. L., Löfverström, M., Lora, J. M., Mikolajewicz, U., ... & Chandan, D. (2020). The PMIP4-CMIP6 Last Glacial Maximum experiments: preliminary results and comparison with the PMIP3-CMIP5 simulations. *Climate of the Past*.

Khodri, M., Leclainche, Y., Ramstein, G., Braconnot, P., Marti, O., & Cortijo, E. (2001). Simulating the amplification of orbital forcing by ocean feedbacks in the last glaciation. *Nature*, *410*(6828), 570-574.

Nolan, C., Overpeck, J. T., Allen, J. R., Anderson, P. M., Betancourt, J. L., Binney, H. A., ... & Djamali, M. (2018). Past and future global transformation of terrestrial ecosystems under climate change. *Science*, *361*(6405), 920-923.

Robock, A., Ammann, C. M., Oman, L., Shindell, D., Levis, S., & Stenchikov, G. (2009). Did the Toba volcanic eruption of~ 74 ka BP produce widespread glaciation?. *Journal of Geophysical Research: Atmospheres*, *114*(D10).

Robock, A., Oman, L., & Stenchikov, G. L. (2007). Nuclear winter revisited with a modern climate model and current nuclear arsenals: Still catastrophic consequences. *Journal of Geophysical Research: Atmospheres*, *112*(D13).

Xu, C., Kohler, T. A., Lenton, T. M., Svenning, J. C., & Scheffer, M. (2020). Future of the human climate niche. *Proceedings of the National Academy of Sciences*, *117*(21), 11350-11355.

---

## Short Comment (SC2) · 4 Jun 2020

Thank you for your response. Your explanations are helpful, and the modifications to the text you propose will definitely improve it.

The only concern I still have relates to your following statement: "We make the hypothesis of a return to the LGM, even if physically implausible, and merely have a look at what the consequences would be according to econometrics. Our main focus is on highlighting the unrealistic results obtained with statistical damage functions for a climatic change symmetrical to the RCP8.5 (when looking only at the mean temperature), not to discuss the physical mechanisms that could trigger such a change."

[Figure]

Frankly, I have a difficulty to imagine a situation where physically implausible arguments would have any value. If the LGM is implausible in 2100, then all your arguments regarding the economic impact are also implausible. How can we judge that the statistical damage function is unrealistic if you compare it with unrealistic world? Again, the absurdity must be measured against physical laws. The econometrics you challenge may be absurd (unrealistic) only if physics tells us that, for example, in 2100 North America and Europe will be covered by ice. But if it is not physically plausible, then the econometrics seems to be valid. If physically implausible arguments are admitted, one may come up with multitudes of equally implausible scenarios that support the econometrics you contest.

Therefore, plausibility of a scenario should always be a concern. Let us estimate it from very general scaling considerations. The empirical energy density spectrum of Huybers and Curry (2006) has a frequency slope of roughly $B \approx 1.64$ in northern latitudes. Since the energy density slope B relates to the fluctuation amplitude slope b as B=2b+1, $B \approx 1.64$ corresponds to b = 0.32. Therefore, the amplitude of the climate system response to 0.1-kyr forcing relates to the amplitude of the 100-kyr response as $10^{(-0.96)} = 0.11$. Thus, regardless of the physical nature of the centennial forcing you want to invoke for your scenario, in 2100 you may count at best on ~10% response relative to LGM. Perhaps it is enough to make a case regarding the validity of the econometrics. Otherwise, to discredit the econometrics, one needs to come up with a justification of the centennial forcing amplitude which is 10 times stronger than it was observed in the past.

Reference: Huybers, P. and Curry, W.: Links between annual, Milankovitch and continuum temperature variability, Nature, 441, 329–332, https://doi.org/10.1038/nature04745, 2006.

---

## Author Comment (AC2) · 8 Jun 2020

One of the issues concerning potential future climate change is that there is no analogue in the "recent" past of our planet of such a change. There are similar changes in magnitude, but not in rate. As pointed out by Nolan et al. (2018) in their study, "under the RCP8.5 scenario the rate of warming will be on the order of 65 times as high as the average warming during the last deglaciation". Therefore, to illustrate the disconnect between climate sciences and econometrics, two options can be considered:

1) Carefully list all the expected climate and environmental changes according to climate models at the end of the century (including extreme events, sea level rise...etc)

that are not accounted for by statistical damage functions and show that they would have an impact far above the projections from these functions. This was the approach of DeFries et al. (2019). This option relies on current Earth system models, and there are still many uncertainties.

2) Apply these functions to a different, but rather well-known, climate change that has occurred in the past. We chose the LGM because it is the most recent past period representing a climate change of the same magnitude than what may be our future (RCP8.5). This period is actually often used in climate communication to illustrate the fact that a difference of 4°C in the global mean temperature is by no mean a small change but corresponds to a completely different world. As mentioned previously, the comparison between the LGM and the RCP8.5 has already been made by Nolan et al. (2018) to assess ecosystem changes.

On the one hand, going back to exactly the same climate state than the LGM would require following exactly the same path than for the last glacial period, with the same forcings. By definition, this is not possible (even the different glacial periods of the Pleistocene have their own climate and ice-sheet patterns). In this regard, our scenario is implausible.

But on the other hand, the glacial climate state itself is not physically implausible, since it has already occurred in the past. We excluded the ice sheets from our equilibrium assumption, because considering that the LGM extent and thickness of the Lauren- tide and Fennoscandian ice sheets is reached would be like assuming more than 20 m of sea level rise in 2100 for the RCP8.5, by comparison with the mid-Pliocene esti- mates. Looking at the surface mass balance over the Laurentide and Fennoscandian ice sheets at the LGM, as simulated with the IPSL_CM4 climate model, it appears that the balance is positive over most of the ice sheets (except on the edges), with val- ues above 40 cm/year on the southern edges (outside the ablation zone) and 10-20 cm/year more in the center (Woillez et al., 2012). But the spatial resolution of this sim- ulation is rather coarse. Other simulations from Ullman et al. (2015) for the Laurentide

ice sheet actually show that the accumulation rate is above 50 cm/year on the edges, but very low in the center, as you suggested it could be because of the drier glacial climate.

Based on these elements, I consider that, in our thought experiment, snow accumulation on the regions corresponding to the edges of the LGM northern ice sheets would be of a few tens of cm/year in the last decades of the scenario, which would lead to an accumulation of a few meters of snow. The total thickness would of course depend on when the threshold for accumulation is crossed, depending on temperature and precipitation evolution. In the more central regions, it is difficult to provide an estimate based only on published surface-balance maps for the LGM. Yet, for these regions, the decrease of the mean annual temperature is greater than 20°C, which makes them rather unsuitable for significant human activities anyway.

To summarize, a better assessment of what snow accumulation would be at the end of the century would require performing ad hoc simulations using our hypothetical climate scenario as an input for a surface model including a snow model. But we can consider that on the areas corresponding to the LGM ice sheets there would be either a permanent snow layer of a few meters and/or that the climate would be much too cold for human activities. In both cases, we cannot expect the current level of economic activities to be maintained, even less so to grow. Therefore, we do not think that this issue of ice sheets invalidates our demonstration, but we suggest that 1) the above arguments should be added in the manuscript; 2) given the uncertainties on the snow accumulation rate without new simulations we cannot constrain the rate of sea level drop and whether it would be fast enough to really have a significant economic impact, therefore it might be better not to consider these impacts. Moreover, the inconsistency of the GDP projections for southern countries remain: if the damage functions of Burke et al. (2015) manage to simulate a collapse of northern countries because of cold temperatures, the results obtained for southern countries (large positive impacts on GDP) appears unrealistic compared to the climate and ecosystem changes (sahelian

countries for instance).

References:

Woillez, M. N., Krinner, G., Kageyama, M., & Delaygue, G. (2012). Impact of solar forcing on the surface mass balance of northern ice sheets for glacial conditions. Earth and Planetary Science Letters, 335, 18-24.

Ullman, D. J., Carlson, A. E., Anslow, F. S., LeGrande, A. N., & Licciardi, J. M. (2015). Laurentide ice-sheet instability during the last deglaciation. Nature Geoscience, 8(7), 534-537.

---

## Referee Comment (RC1) · Anonymous Referee #1 · 18 Jun 2020

This is a peculiar paper.

The authors take two functions for the economic impact of weather and pretend that these represent the economic impact of climate change. One of these functions is unpublished, the other is known to be wrong.

The authors estimate these functions for a period of modest warming and extrapolate to a scenario with large cooling. They pay little attention to specification and confidence intervals.

The authors refer to but do not use the functions of the economic impact of climate change. They ignore the one study of the economic impact of cooling by Ralph d'Arge.

They do not compare their results to previous estimate of the impact of a shutdown of the thermohaline circulation, the scenario that comes closest to the one considered here.

I recommend a major revision because the idea is nice, but major revision here means replace the paper.

Equation (1) is wrong. The left-hand side is stationary. Temperature on the right-hand side is non-stationary. This cannot be, but fortunately there are year dummies. You thus regress economic growth on the cointegrating vector of temperature and trend. The statistical properties of this are unknown, but results are biased because you measure the cointegrating vector with error. The procedure may work in sample, but you cannot extrapolate without estimating the year dummies for future years. Burke set future dummies to zero, which is really quite a stupid thing to do.

---

## Referee Comment (RC2) · Anonymous Referee #2 · 21 Jun 2020

June 21, 2020

This is an interesting article. While I see the point (raised by an earlier reviewer and editor) that it does not generate much knowledge on the real (warming) world, I believe it neatly summarises and illustrates issues with climate damage function, with an original twist of argumentation. The current article could stimulate discussion on the merits, limitations, and validation of damage functions, which would be a valuable contribution to the scientific discourse around climate change.

I therefore think that the article should be published. However, there are some issues which require clarification, as listed below.

Major Comments

Inconsistency: "known" cooling vs "unknown" warming scenario

An important line of argumentation seems to me that while we don't know what a warming world looks like, we can form an idea about a cooling of similar magnitude by looking at the ice age data. However, in fact, the paper does not assume a full transition to an ice age (which would involve long-term equilibration of ice sheets etc) but a quick cooling, i.e. a scenario for which we have no data. The uncertainty may involve the question of snow accumulation, raised by an earlier interactive comment, and effects

depending on it (e.g. circulation changes due to the albedo effect of the snow), but also changes in ocean circulations (how does the AMOC react to the cooling?). The fast cooling scenario may thus differ form an ice age in more respects than the presence or absence of ice caps.

So, is the damage in a fast cooling scenario just as speculative and difficult to assess than in a warming scenario?

There are in fact two types of uncertainty here, 1. what would the state of the climate system be like under fast cooling, 2., what would be the impacts for society?

In my view, there are two ways to deal with the first issue.

- Simply *define* that your cooling scenario is "an ice age except that ice sheets are not there yet". You would have to explicitly acknowledge that this may not be the actual response of the Earth system to a cooling stimulus (such as rapid greenhouse gas depletion), but you could still analyse the potential (societal) impact of such a hypothetical climate.

- Or perform a model simulation (/team up with a modelling group) of a 4degree cooling in 100 years, either by dropping GHG concentrations or a reduction in solar irradiation.

I strongly encourage you to consider the second option.

Once you clarified your climate scenario, you can argue, as you do now, that the impact of society would be severe (with higher certainty than the severity of an equivalent warming?).

Asymmetry warming - cooling

The ice-age scenario obviously contains very severe impacts for human activities, many of which cannot be captured by looking at recent data, as the ice-age Earth might be a qualitatively different place from our current world.

However, this does not *automatically imply* that a warming of equal magnitude would have similarly huge impacts.

This does not invalidate your main argument, that (statistical) damage functions *may* well overlook major impacts of climate change which current data cannot capture, but I would like this asymmetry to be acknowledged explicitly. Even better, if possible, would be to include a brief discussion on whether it is plausible/impossible to know/implausible that warming has similarly strong impacts as cooling. For example, how does the area (or number of inhabitants, or value of infrastructure) threatened by sea level rise under 4 degree warming compare to the area (or number of people/amount of wealth) threatened by snow under 4 degree cooling? Obviously, uncertainties are huge, but maybe something meaningful can still be said about the issue?

Enumerative vs. data-driven damage function

You use two damage functions of the statistical kind and none of the enumerative kind. Is this a conscious choice, and could you please motivate it? For example, did you make this choice because statistical damage functions inherently include the (statistical) effect of both cooling and warming, whereas the enumerative ones primarily look at warming (e.g. they may include a term for heat stress on maize plants, but not for frost stress...)?

In particular, it seems to me that your argumentation shows that capturing climate damage exhaustively with a statistical approach is impossible, whereas an enumerative

approach could work in principle (but maybe not in practice). Please clarify.

Minor Comments

- Line 253: "these regions would be about as suitable for humans as present-day Arctic is"... Instead of this picturesque metaphor, I suggest to specify the conditions (how cold and dry? unsuitable for any form of present-day agriculture, forestry, even Sami-style animal husbandry?).

- Line 266ff: "Most places would become unsuitable for agriculture and water resources would largely decrease. Drier regions include ...India and Indonesia". Would drying be a severe concern in regions that are currently wet (like Indonesia and parts of India)? And even if rainfall decreases, could it not be that the reduction of evaporation due to cooling compensates the effect, leading to no severe increase in drought? Note that several regions, including the Mediterranean, and parts of South Africa, are threatened by drought under global warming (for example because of poleward expansion of the ITCZ system and hence the subtropical deserts). Of course, drought needn't be linear in global mean temperature, but possibly these regions would get less drought-prone under global cooling.

- This reference could be interesting for the general discussion on damage functions: JCV Pezzey, "Why the social cost of carbon will always be disputed", https://onlinelibrary.wiley.com/doi/full/10.1002/wcc.558,

Technical Comments (typos etc)

- line 265: o fthe -> of the

- line 294: does not captures -> capture

- fig. 1: Islande -> Iceland

---

## Referee Comment (RC3) · Anonymous Referee #1 · 22 Jun 2020

You're supposed to respond to comments, rather than dismiss them.

---

## Author Comment (AC3) · 22 Jun 2020

***The comments of referee#1 are reproduced below in black. Our responses are in blue.***

This is a peculiar paper.

The authors take two functions for the economic impact of weather and pretend that these represent the economic impact of climate change. One of these functions is unpublished, the other is known to be wrong.

- We believe such functions based on economic impact of weather cannot represent the economic impact of climate change. However, the authors of these functions (Burke et al. and Newell et al.) actually do. All the idea in our work is to illustrate to what absurd results using recent past weather to estimate large climate change impact can lead.
- The function of Burke et al. (2015) has been published in Nature, has been cited in the literature several hundred times and has been used to compute the social cost of carbon (e.g. Ricke et al., 2018). The authors also published several other papers based on similar methodologies (e.g. Hsiang (2016), Burke et al. (2018), Diffenbaugh & Burke (2019)). In our opinion, if the function is known to be wrong this point has not received enough attention so far, at least among economists.
- The function of Newell et al. (2018) has not been published in a peer-reviewed journal, but it was interesting to consider it anyway because 1) it belongs to the family of damage functions assuming an impact of climate on GDP level rather than growth, leading to very small damages; 2) it is based on the same data and methodology than Burke et al. (2015).

Burke, M., Davis, W. M., & Diffenbaugh, N. S. (2018). Large potential reduction in economic damages under UN mitigation targets. *Nature*, *557*(7706), 549-553.

Diffenbaugh, N. S., & Burke, M. (2019). Global warming has increased global economic inequality. *Proceedings of the National Academy of Sciences*, *116*(20), 9808-9813.

Hsiang, S. (2016). Climate econometrics. *Annual Review of Resource Economics*, *8*, 43-75.

Ricke, K., Drouet, L., Caldeira, K., & Tavoni, M. (2018). Country-level social cost of carbon. *Nature Climate Change*, *8*(10), 895-900.

The authors estimate these functions for a period of modest warming and extrapolate to a scenario with large cooling. They pay little attention to specification and confidence intervals.

- We perform an "ad absurdum" demonstration. Therefore, we consider that using only the main specification of Burke et al. (2015), which is the most analyzed in their work, is sufficient for the sake of the demonstration.

The authors refer to but do not use the functions of the economic impact of climate change. They ignore the one study of the economic impact of cooling by Ralph d'Arge.

- I do not fully understand your remark. We refer to other functions in the introduction, as a general context, but our work did not aim at applying all published damage functions to our hypothetical cooling.
- I assume you refer to the paper of Ralph d'Arge from 1979, "Climate and economic activity". We did not cite this paper because it is cited in the review of Tol (2018), which is mentioned in the introduction. Indeed, d'Arge (1979) estimated the impact of a 1°C cooling, and found a very small impact of -0.6% in the world GDP. So, as it

seems to be the only previous paper investigating the impact of a cooling we could add it in the references, but the magnitude of cooling is anyway much smaller than the one we assume in our work. Also, d'Arge's work does not seem to be available on the internet, therefore I do not know what was his methodology, but considering the large development in climate sciences over the past 40 years, this work might be out of date.

They do not compare their results to previous estimate of the impact of a shutdown of the thermohaline circulation, the scenario that comes closest to the one considered here.

- Concerning the impact of a shutdown of the THC, there is indeed for instance the work by Link & Tol (2011), using the FUND model. But their work uses an integrated assessment model, when we restricted ourselves to econometrics methods. Also, in their work, the cooling induced by the collapse of the THC is much smaller than in our work (only -1,7°C in average on the north hemisphere) and occurs within the context of global warming. The scenario is therefore utterly different from ours.
- There are also the works of Kuhlbrodt et al. (2009) or Anthoff et al. (2016), but again it uses integrated assessment models, and the THC collapse occurs within the context of global warming, so that the global temperature change is much smaller than in our scenario. Moreover, in the case of THC collapse the cooling is more or less limited to the north hemisphere.
- These works rely on climate projections from numerical modelling only, with all the associated uncertainties, whereas we wanted to refer to a known past period for which there are also data, as explained in our responses to M. Verbitsky.

Anthoff, D., Estrada, F., & Tol, R. S. (2016). Shutting down the thermohaline circulation. *American Economic Review*, *106*(5), 602-06.

Kuhlbrodt, T., Rahmstorf, S., Zickfeld, K., Vikebø, F. B., Sundby, S., Hofmann, M., ... & Jaeger, C. (2009). An integrated assessment of changes in the thermohaline circulation. *Climatic Change*, *96*(4), 489-537.

Link, P. M., & Tol, R. S. (2011). Estimation of the economic impact of temperature changes induced by a shutdown of the thermohaline circulation: an application of FUND. *Climatic Change*, *104*(2), 287-304.

I recommend a major revision because the idea is nice, but major revision here means replace the paper.

Equation (1) is wrong. The left-hand side is stationary. Temperature on the right-hand side is non-stationary. This cannot be, but fortunately there are year dummies. You thus regress economic growth on the cointegrating vector of temperature and trend. The statistical properties of this are unknown, but results are biased because you measure the cointegrating vector with error. The procedure may work in sample, but you cannot extrapolate without estimating the year dummies for future years. Burke set future dummies to zero, which is really quite a stupid thing to do.

- We strictly followed Burke et al. (2015) equations. The criticism you put forward is interesting and to be honest we never thought about it. We would hence appreciate if you could provide some references where this has been published. Discussing the mathematical issues of Burke et al. approach would be very useful and certainly strengthen our argument. That being said, our point was not to engage into that line of criticism but rather to merely do an "ad-absurdum" demonstration. We indeed

believe, as stressed in the response to your first point, that, given the wide coverage and impact of the Burke et al. (2015) and ensuing papers, a more explicit demonstration of the irrelevance of the approach is a useful complementary contribution to a more mathematical/statistical critique.

---

## Referee Comment (RC4) · Anonymous Referee #2 · 8 Jul 2020

Thanks for your thorough reply. If you implement the outlined changes in the paper, I expect I will be satisfied. Looking forward to seeing the final version.

---

## Author Comment (AC4) · 8 Jul 2020

***The comments of referee#2 are reproduced below in black. Our responses are in blue.***

This is an interesting article. While I see the point (raised by an earlier reviewer and editor) that it does not generate much knowledge on the real (warming) world, I believe it neatly summarises and illustrates issues with climate damage function, with an original twist of argumentation. The current article could stimulate discussion on the merits, limitations, and validation of damage functions, which would be a valuable contribution to the scientific discourse around climate change.

I therefore think that the article should be published. However, there are some issues which require clarification, as listed below.

> ➢ Thank you for your positive comments on our work. Indeed, our main goal is not to generate knew knowledge on future climate change and its impacts but to question the relevance of (some) current damage functions.

Major Comments

Inconsistency: "known" cooling vs "unknown" warming scenario

An important line of argumentation seems to me that while we don't know what a warming world looks like, we can form an idea about a cooling of similar magnitude by looking at the ice age data. However, in fact, the paper does not assume a full transition to an ice age (which would involve long-term equilibration of ice sheets etc) but a quick cooling, i.e. a scenario for which we have no data. The uncertainty may involve the question of snow accumulation, raised by an earlier interactive comment, and effects depending on it (e.g. circulation changes due to the albedo effect of the snow), but also changes in ocean circulations (how does the AMOC react to the cooling?). The fast cooling scenario may thus differ form an ice age in more respects than the presence or absence of ice caps.
So, is the damage in a fast cooling scenario just as speculative and difficult to assess than in a warming scenario?

There are in fact two types of uncertainty here, 1. what would the state of the climate system be like under fast cooling, 2., what would be the impacts for society?

In my view, there are two ways to deal with the first issue.

• Simply define that your cooling scenario is "an ice age except that ice sheets are not there yet". You would have to explicitly acknowledge that this may not be the actual response of the Earth system to a cooling stimulus (such as rapid greenhouse gas depletion), but you could still analyse the potential (societal) impact of such a hypothetical climate.
• Or perform a model simulation (/team up with a modelling group) of a 4degree cooling in 100 years, either by dropping GHG concentrations or a reduction in solar irradiation.

I strongly encourage you to consider the second option.

Once you clarified your climate scenario, you can argue, as you do now, that the impact of society would be severe (with higher certainty than the severity of an equivalent warming?).

> ➢ Your comment is quite similar to the previous interactive comment of M. Verbitsky and we agree that we should clarify our climate change hypothesis: we assume a return to the last glacial maximum, except for the ice sheets. The hypothesis includes ecosystem changes, which were triggered not only by temperature and precipitation changes but also by lower CO2. Of course, this hypothesis implies some inconsistency, as the LGM climate takes into account the albedo and elevation impact

of the ice sheets. We acknowledge that the response of the Earth system to a forcing that would lead to a 4°C cooling in 2100 would be different from the LGM (depending on the type of forcing) and therefore the LGM is not a perfect reverse analog of a future at +4°C. A short discussion on that issue will be added to the manuscript. Please also refer to the response to M. Verbitsky for the other modifications that would be implemented in our manuscript to clarify this issue.

Performing some ad hoc climate modelling to simulate a cooling of 4°C by 2100 would indeed allow avoiding the above mentioned issue. However, it would, in our view, raise at least two new issues:

1) Unrealistic forcing mechanism: As you mentioned, the forcing mechanism could be either a strong reduction in solar irradiation or a drop in GHG concentrations. But to reach -4°C, the decrease in solar irradiation would have to be much stronger than the natural changes currently reconstructed for the past millennia and a decrease of GHG would have to be even larger than for the LGM (since GHG drop alone during the LGM is not sufficient to simulate a full glacial climate, e.g. Kim, 2004.), which could not be reached within a century with natural mechanisms. Therefore, we would have to assume some anthropogenic factors, like massive atmospheric $CO_2$ pumping and storage, which would be unrealistic. The last option would be a massive volcanic eruption, but it would have to last continuously for several decades, which would be a very questionable hypothesis.

2) Most importantly, the climate scenario would then rely on climate modelling only, with only one model, with all the associated uncertainties, especially concerning ecosystem changes. Therefore, confronting the climate projections with the damage projections would not be different from doing the same exercise in the case of a warming scenario (which of course remains a valid option to illustrate the inconsistency of climate damages projections). In our view, the main interest of using the LGM as a benchmark to test econometric models based on mean annual temperature is to have not only climate simulations but also various proxy data on the climate and ecosystem at that time.

This is why we decided to stick with the first proposed solution of clarifying our climate change hypothesis.

Asymmetry warming – cooling

The ice-age scenario obviously contains very severe impacts for human activities, many of which cannot be captured by looking at recent data, as the ice-age Earth might be a qualitatively different place from our current world.

However, this does not automatically imply that a warming of equal magnitude would have similarly huge impacts.

This does not invalidate your main argument, that (statistical) damage functions may well overlook major impacts of climate change which current data cannot capture, but I would like this asymmetry to be acknowledged explicitly. Even better, if possible, would be to include a brief discussion on whether it is plausible/impossible to know/implausible that warming has similarly strong impacts as cooling. For example, how does the area (or number of inhabitants, or value of infrastructure) threatened by sea level rise under 4 degree warming compare to the area (or number of people/ amount of wealth) threatened by snow under 4 degree cooling? Obviously, uncertainties are huge, but maybe something meaningful can still be said about the issue?

We agree that this point should be acknowledged explicitly.

When looking at some of the most dramatic climate projections for RCP8.5 at the end of the century, it seems rather plausible that warming would have similar strong impacts as a cooling of similar magnitude:

- On the one hand, at the LGM, large parts of North America and Europe would become rather unsuitable for large human populations and most modern economic activities. Currently 37 millions of people live in Canada and about 30 millions in northern Europe, where we can reasonably assume that only a small population could remain. Maintaining a total population of more than 700 millions of people in Europe despite the very cold temperature in winter and permafrost expansion is doubtful, even if the number of people that could still live there (probably mostly in southern Europe) remains speculative. Similarly, in India, data suggests that the north-western part of the continent experienced extreme desert conditions during the LGM (Ansari et al., 2007), which would probably have strong negative impact for its current 70 millions of inhabitants.

- On the other hand, the number of people currently living in areas which may be exposed to permanent inundations for a sea level rise of 1,46 m in 2100 was recently estimated to 340 millions (population growth and migration not taken into account, Kulp & Strauss 2019), a number that would be further increased for higher SLR values (SLR>2,4m has 5% probability according to Bamber et al., 2019). But the most alarming projections are maybe the ones concerning future heat stress: according to Mora et al. (2017), temperature and humidity conditions above potentially deadly threshold could occur nearly year-round in humid tropical areas, including some of the most densely populated areas, threatening hundreds millions of people. How people could adapt to such unprecedented climatic conditions remains an open question.

However, while we agree that from a physical point of view there is no a priori reason to postulate that warming and cooling of similar magnitude would have similar huge impacts, it should be noticed that this issue of symmetry between a warming and a cooling is implicitly assumed by the damage function itself: it was built on both negative or positive temperature anomalies. Therefore, by design, it cannot be assumed that such function would provide relevant damage estimates for a warming but not for a cooling (or the reverse).

A paragraph explaining the above points will be added in the general discussion.

Enumerative vs. data-driven damage function

You use two damage functions of the statistical kind and none of the enumerative kind.
Is this a conscious choice, and could you please motivate it? For example, did you make this choice because statistical damage functions inherently include the (statistical) effect of both cooling and warming, whereas the enumerative ones primarily look at warming (e.g. they may include a term for heat stress on maize plants, but not for frost stress...)?

In particular, it seems to me that your argumentation shows that capturing climate damage exhaustively with a statistical approach is impossible, whereas an enumerative approach could work in principle (but maybe not in practice). Please clarify.

➢ Yes, this is a conscious choice, which will be justified briefly in section 2. As you pointed it out, enumerative damage functions are designed for a warming and we could not have applied them to a cooling. They may lead to implausible results as well (especially at a global scale), because some impacts are not accounted for or because the sectorial impacts are generalized based on evidence limited to a short time period or small spatial scale for instance. In that case, illustrating the disconnect

with climate sciences would have required a different approach (e.g. questioning their assumptions, sector by sector, or providing damage estimates for impacts usually not taken into account, like extreme events).
➢ We consider that the enumerative approach could work in principle, providing that the potential cascading effects could be taken into account, but we doubt that this could actually be done at the global scale. Damage projections may be possible, to some extent, at region or country scale, but it remains a complex and challenging work and it is highly probable that high levels of uncertainties will remain, as very well pointed out in the article of Pezzey that you have indicated. A short paragraph will be added in the conclusion, to further discuss this point, including some of the issues raised by Pezzey (2017).

Minor Comments

• Line 253: "these regions would be about as suitable for humans as present day Arctic is"... Instead of this picturesque metaphor, I suggest to specify the conditions (how cold and dry? unsuitable for any form of present-day agriculture, forestry, even Sami-style animal husbandry?).

Please find below in red some additional information, which will be added to the text:

"In Europe, the mean annual temperature would decrease by 4-8°C in the Mediterranean region, by 8-12°C over the western, central and eastern regions and by more than 12°C over northern countries (Fig.2). For France for instance, whose current mean annual temperature is about 11°C, the temperature decrease would thus correspond to a shift to the current mean temperature of northern Finland. Over Western Europe, the mean temperature of the coldest month would decrease by 10-20°C (Ramstein et al., 2007) and mean annual precipitation would decrease by about 300 mm/year (Wu et al., 2007).   Forests would be highly fragmented, replaced by steppe or tundra vegetation (Prentice et al., 2011). The southern limit of the permafrost would approximately reach 45°N, i.e. the latitude of Bordeaux (Vandenberghe et al., 2014). In such a context, maintaining European agriculture at its current state, among other human activities, would be a costly and technically highly demanding challenge. Energy needs for heating would tremendously increase, current infrastructures would be damaged by severe frost and it is doubtful that Europe could still sustain its current population on lands preserved from permanent snow accumulation. In Asia, similar problems would occur, with a decrease in mean annual temperature between 4 to 8°C over most Chinese regions for instance (Fig.1) The boreal forest would progressively vanish, replaced by steppe and tundra (Prentice et al.,2011). Permafrost would extend in the North-East and North China, up to Beijing, as well as in the west of the Sichuan (Zhao et al., 2014). As a result, rice cultivation in the northern province of Heilongjiang, currently >20.10[6] tons/year, i.e. about 10% of the national production (Clauss et al. (2016), would no longer be possible.  Permafrost would not stretch out to the whole densely populated North China plains, but the cold and dry climate there would nonetheless prevent rice cultivation. The discharge of the Yangtze River at Nanjing would be less than half its present-day value (Cao et al., 2010), questioning current hydroelectricity production. In short, current livelihoods in these regions would no longer be sustainable and population would probably be much lower than today. "

• Line 266ff: "Most places would become unsuitable for agriculture and water resources would largely decrease. Drier regions include ...India and Indonesia".

Would drying be a severe concern in regions that are currently wet (like Indonesia and parts of India)? And even if rainfall decreases, could it not be that the reduction of evaporation due to cooling compensates the effect, leading to no severe increase in drought? Note that several regions, including the Mediterranean, and parts of South Africa, are threatened by drought under global warming (for example because of poleward expansion of the ITCZ system and hence the subtropical deserts). Of course, drought needn't be linear in global mean temperature, but possibly these regions would get less drought-prone under global cooling.

It is indeed important to take into account temperature, whose decrease could compensate for the precipitation decrease. Compilations of lake levels at the LGM indeed show that some were higher at the LGM whereas other where lower (McGee, 2020) and climate models show that drier places in terms of precipitation were not always drier in terms of hydrology (Scheff, 2017). This point, which has not been discussed in our manuscript, will be added in section 5.

Concerning Indonesia, you are right. Data for that region are rather sparse, and showing spatial variability: for Borneo for instance it seems that the vegetation cover was broadly similar during the Holocene and the LGM, suggesting that there was no pronounced dry season, whereas for Sumba, pollen data suggest enhanced aridity and water stress during the dry season (Dubois et al., 2014). Thus, it seems actually difficult to make hypotheses on the potential impact for human populations and the reference to Indonesia will either be suppressed or the uncertainties will be explained.

For India, data are also rather sparse. Analyses of a marine core in the Bay of Bengale suggest that fluvial discharge was reduced during the LGM, but the decrease is not quantified (Duplessy, 1982). Looking at simulation results, it seems that there was a decrease in P-E (Scheff, 2017) and a large decrease in runoff across monsoonal Asia, including India and south-east Asia (Li et al., 2013), suggesting that the decrease in temperature did not compensate for the decrease in precipitation. This could mean less flooding during the monsoon season, but also a decrease of water resources during the dry season, in areas where droughts are already a problem at present-day. Unfortunately, to our knowledge, there is no publication on the seasonality of runoff during the LGM. In regions already dry today, like the Pakistan or north western India, it seems that conditions were even more arid during the last glacial (Ansari & Vink, 2007). These references will be added in section 5.

• This reference could be interesting for the general discussion on damage functions: JCV Pezzey, "Why the social cost of carbon will always be disputed", https://onlinelibrary.wiley.com/doi/full/10.1002/wcc.558,

  ➢ Thanks for this reference, which will be added to the general discussion on damage functions. It nicely (and sometimes provokingly) summarizes the different issues, including for the statistical approach. It could also be cited in the conclusion, since it questions the very social/political utility of damage functions.

Technical Comments (typos etc)

• line 265: o fthe -> of the
• line 294: does not captures -> capture

• fig. 1: Islande -> Iceland

Additional references:

Ansari, M. H., & Vink, A. (2007). Vegetation history and palaeoclimate of the past 30 kyr in Pakistan as inferred from the palynology of continental margin sediments off the Indus Delta. *Review of Palaeobotany and Palynology*, *145*(3-4), 201-216.

Bamber, J. L., Oppenheimer, M., Kopp, R. E., Aspinall, W. P., & Cooke, R. M. (2019). Ice sheet contributions to future sea-level rise from structured expert judgment. *Proceedings of the National Academy of Sciences*, *116*(23), 11195-11200.

Cao, G., Wang, J., Wang, L., & Li, Y. (2010). Characteristics and runoff volume of the Yangtze River paleo-valley at Nanjing reach in the Last Glacial Maximum. *Journal of Geographical Sciences*, *20*(3), 431-440.

Clauss, K., Yan, H., & Kuenzer, C. (2016). Mapping paddy rice in China in 2002, 2005, 2010 and 2014 with MODIS time series. *Remote Sensing*, *8*(5), 434.

Dubois, N., Oppo, D. W., Galy, V. V., Mohtadi, M., Van Der Kaars, S., Tierney, J. E., ... & Linsley, B. K. (2014). Indonesian vegetation response to changes in rainfall seasonality over the past 25,000 years. *Nature Geoscience*, *7*(7), 513-517.

Duplessy, J. C. (1982). Glacial to interglacial contrasts in the northern Indian Ocean. *Nature*, *295*(5849), 494-498.

Kim, S. J. (2004). The effect of atmospheric CO 2 and ice sheet topography on LGM climate. *Climate dynamics*, *22*(6-7), 639-651.

Kulp, S. A., & Strauss, B. H. (2019). New elevation data triple estimates of global vulnerability to sea-level rise and coastal flooding. *Nature communications*, *10*(1), 1-12.

Li, Y., & Morrill, C. (2013). Lake levels in Asia at the Last Glacial Maximum as indicators of hydrologic sensitivity to greenhouse gas concentrations. *Quaternary Science Reviews*, *60*, 1-12.

McGee, D. (2020). Glacial–Interglacial Precipitation Changes. *Annual Review of Marine Science*, *12*, 525-557

Mora, C., Dousset, B., Caldwell, I. R., Powell, F. E., Geronimo, R. C., Bielecki, C. R., ... & Lucas, M. P. (2017). Global risk of deadly heat. *Nature climate change*, *7*(7), 501-506.

Scheff, J., Seager, R., Liu, H., & Coats, S. (2017). Are glacials dry? Consequences for paleoclimatology and for greenhouse warming. *Journal of Climate*, *30*(17), 6593-6609.

---

## Author Comment (AC5) · 13 Jul 2020

Thank you for your comments, we hope the final revised version will meet your expectations.

---

## Author Comment (AC6) · 23 Jul 2020

In order to respond to some of the points raised here above, please find below the modifications to the manuscript we propose:

- In addition to the changes in the introduction proposed in response to referee#2 to explain why we chose statistical functions and did not consider enumerative ones, we will add the following paragraph in the introduction:

"Different functional forms of statistical damage functions are available in the literature. The choice of Burke et al. (2015) and Newell et al. (2018) was based on the following

considerations:

i) While there might be controversies related to the specification of the model, interpretation and statistical significance of results or even validity of the approach, the approach is well published in leading peer-reviewed journals. The function of Burke et al. (2015) has been cited in the literature several hundred times and has been used to compute the social cost of carbon (e.g. Ricke et al., 2018). The authors also published several other papers based on similar methodologies (e.g. Hsiang (2016), Burke et al. (2018), Diffenbaugh & Burke (2019)).

ii) The function of Newell et al. (2018) has not been published in a peer-reviewed journal, but we considered it anyway because: 1) It belongs to the family of damage functions assuming an impact of climate on GDP level rather than growth, leading to very small damages; 2) It is based on the same data and methodology than Burke et al. (2015), hence simplifying the exercise.

We are well aware of the statistical limitations of the approach, see Keen (2020, in prep.) for a recent summary of these critiques. We nonetheless decided to perform an "ad-absurdum" demonstration. We indeed believe that a more explicit demonstration of the irrelevance of the approach is a useful complementary contribution to a more mathematical/statistical critique. As documented by DeCanio (2003) for older functional forms, the literature on damage functions has had tremendous political implication, and even found its way in IPCC reports. As Keen (2020, in prep), we believe it is important to expose the limitation of the approach."

- Concerning the issue of confidence intervals, we propose to add the following sentence to section 3, L125: "For the simplicity of the demonstration, we chose to consider only one functional form linking temperature to GDP from BHM and NPS: we use either the BHM formula with their main specification, which is the most analyzed in their work [...]".

References

Burke, M., Davis, W. M., & Diffenbaugh, N. S. (2018). Large potential reduction in economic damages under UN mitigation targets. Nature, 557(7706), 549-553.

DeCanio, S. (2003). Economic models of climate change: a critique. Springer.

Diffenbaugh, N. S., & Burke, M. (2019). Global warming has increased global economic inequality. Proceedings of the National Academy of Sciences, 116(20), 9808-9813.

Hsiang, S. (2016). Climate econometrics. Annual Review of Resource Economics, 8, 43-75.

Keen, S. (2020, in prep.). The Appallingly Bad Neoclassical Economics of Climate Change, https://www.patreon.com/posts/appallingly-bad-38048063

Ricke, K., Drouet, L., Caldeira, K., & Tavoni, M. (2018). Country-level social cost of carbon. Nature Climate Change, 8(10), 895-900.